# TransAgent: Transfer Vision-Language Foundation Models with Heterogeneous Agent Collaboration

Yiwei Guo[1,2]  Shaobin Zhuang[*3,4]  Kunchang Li[*1,2,3]  Yu Qiao[3]  Yali Wang[†1,3]

[1]Shenzhen Institutes of Advanced Technology, Chinese Academy of Sciences
[2]University of Chinese Academy of Sciences  [3]Shanghai AI Laboratory
[4]Shanghai Jiao Tong University

## Abstract

Vision-language foundation models (such as CLIP) have recently shown their power in transfer learning, owing to large-scale image-text pre-training. However, target domain data in the downstream tasks can be highly different from the pre-training phase, which makes it hard for such a single model to generalize well. Alternatively, there exists a wide range of expert models that contain diversified vision and/or language knowledge pre-trained on different modalities, tasks, networks, and datasets. Unfortunately, these models are "isolated agents" with heterogeneous structures, and how to integrate their knowledge for generalizing CLIP-like models has not been fully explored. To bridge this gap, we propose a general and concise TransAgent framework, which transports the knowledge of the isolated agents in a unified manner, and effectively guides CLIP to generalize with multi-source knowledge distillation. With such a distinct framework, we flexibly collaborate with 11 heterogeneous agents to empower vision-language foundation models, without further cost in the inference phase. Finally, our TransAgent achieves state-of-the-art performance on 11 visual recognition datasets. Under the same low-shot setting, it outperforms the popular CoOp with around 10% on average, and 20% on EuroSAT which contains large domain shifts. The code will be released at `https://github.com/markywg/transagent`.

## 1  Introduction

Recently, Vision-Language (V-L) foundation models are mainly pre-trained by contrastive learning with massive image-text pairs from web [61, 38, 15]. As a result, they show the potential on a number of downstream visual recognition tasks, by transferring their representations with prompt learning [87, 86] and/or model adaptation [31, 80]. However, target domain data in the open world are diversified, *e.g.*, EuroSAT [35] refers to satellite images that are highly different from web images in the pre-training. With this large domain shift, it is challenging to achieve good generalization only by adopting such a single model (*e.g.*, CLIP), especially under low-shot regime. Alternatively, with the fast development in vision and NLP, there arises a wide range of expert models [33, 9, 43, 49, 7, 18, 64, 11, 46, 12] which contain rich knowledge by pre-training on different modalities, tasks, networks, and datasets. Hence, the natural question is, is it possible to integrate such knowledge to boost vision-language foundation models?

To answer this question, we should further analyze the form of knowledge in these models. The simplest form is the model output, which explicitly exhibits what kind of tasks these models can tackle. However, these models have heterogeneous network structures and outputs, making direct knowledge combinations infeasible. Hence, existing works often choose cascades of these models

---

*Interns at Shanghai AI Laboratory.  †Corresponding Author.

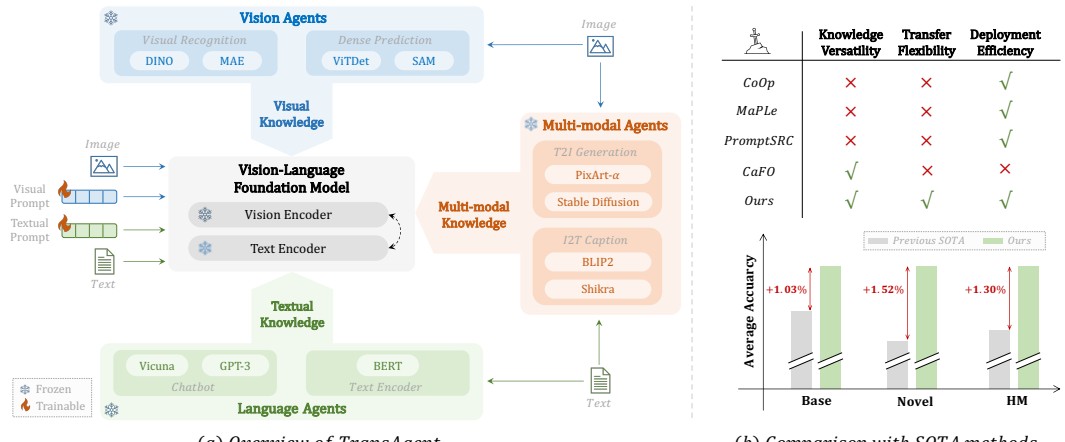

(a) Overview of TransAgent

(b) Comparison with SOTA methods

Figure 1: **An overview of our TransAgent.** (a) TransAgent transfers multi-source knowledge from heterogeneous agents to enhance the generalization ability of vision-language foundation models. It demonstrates knowledge versatility, transfer flexibility and deployment efficiency through elaborate agent collaboration and knowledge ensemble strategy. (b) SOTA comparison for base-to-novel generalization on 11 visual recognition benchmarks. Our method outperforms previous SOTA, especially on the more diversified target domains.

[81, 72, 47], according to their output forms. Apparently, such a design lacks flexibility in transfer learning, based on tool invocation in sequence. Moreover, the resulting pipeline is unfriendly for deployment, due to the ensemble of various models in the inference phase.

Another form is the latent representation which implicitly encodes data knowledge in these models [36, 4, 1]. Compared to the explicit output, this implicit representation has a key advantage, *i.e.*, it is the feature vector that has a homogeneous form among different models. In other words, such vectorized knowledge opens the possibility for a unified integration of these heterogeneous agents. Based on this observation, we propose a general **TransAgent** framework in Figure 1. To our best knowledge, it is the *first* unified distillation framework for generalizing vision-language foundation models with *efficient* heterogeneous agent collaboration. Compared to the previous works mentioned above, our TransAgent contains three distinct technical contributions.

(1) **Knowledge Versatility**. In our TransAgent, we leverage 11 heterogeneous agents from vision, language and multi-modal research, which comprehensively covers diversified knowledge that is complementary with CLIP-like models, from visual recognition to dense prediction, from chatbot to text encoder, from multi-modal generation to caption.

(2) **Transfer Flexibility**. First, we provide a generic knowledge extraction method for each modality, allowing us to flexibly extend more agents if necessary in the future. Especially for multi-modal agents, we design a novel manner to extract the prediction score vector of classes as multi-modal knowledge in the target domain, via elaborate mining of vision-language alignment in these models. Second, we introduce a mixture-of-agents gating mechanism for integrating external knowledge of different agents in each modality. This allows our TransAgent to automatically select agents via soft weighting so that it can adaptively tackle few-shot settings in different domains of target datasets.

(3) **Deployment Efficiency**. We leverage multi-source distillation to transfer knowledge of these heterogeneous agents into CLIP. Since all these pre-trained models are frozen, the fine-tuning effort is neglectable with a few learnable prompts. More importantly, we can unload all the external agents after distillation, *i.e.*, the inference pipeline with the enhanced CLIP is just the same as the original one, achieving deployment efficiency without a heavy model ensemble.

Finally, we conduct extensive experiments on 11 visual recognition benchmarks, where our TransAgent achieves the state-of-the-art under the same low-shot transfer setting, *e.g.*, via knowledge collaboration, it outperforms the well-known CoOp [87] with around 10% on average and 20% on EuroSAT which contains large domain shifts. Our method also achieves better results than CaFo [81], which adopts a model ensemble strategy.

## 2 Related Work

**Foundation models.** The rapid advancements in deep learning methods have brought abundant pre-trained models to the research area. We group these models into four categories and further demonstrate their ideas below. **(i) Vision models:** Vision foundation models [33, 9, 3, 59, 32, 13] pre-trained on ImageNet [21] have shown outstanding transfer capability in visual recognition by fine-tuning on downstream datasets. Moreover, various models [43, 49, 16, 17, 8] can be specialists in dense prediction tasks by pre-training on task-relevant domain data. **(ii) Large language models:** The emergence of large language models (LLMs) [7, 18, 69, 70, 27] has been raising increasing attention from the research community and the public. The astonishing comprehension ability of the LLMs is credited to the linguistic knowledge which can be further applied to solve vision tasks [51, 20, 88, 79]. **(iii) Text-to-image generative models:** Text-conditioned generation task requires high-level understanding of the given prompts. Recently, diffusion-based generative models [37, 23, 64, 11, 56, 62, 66] have become the state-of-the-art. These models can follow the text conditions faithfully and generate desired outcomes, owing to the semantic knowledge learned during the pre-training stage. **(iv) Image-to-text captioning models:** These models typically integrate visual knowledge into LLMs to obtain multi-modal understanding abilities [46, 12, 15, 2, 14, 25, 47], offering better experience in referential dialog scenario. In this work, we excavate the underlying knowledge in these heterogeneous models to empower the VL foundation models.

**Few-shot adaptation.** To efficiently transfer vision-language foundation models like CLIP [61] to downstream tasks, researchers have proposed various adaptation methods, which are primarily based on prompt learning [87, 86, 39–41, 65, 85, 53, 50] or adapter [31, 80, 68, 48, 81, 75, 89]. Lu et al. [53] explore the potential of collaborating CLIP's architectural variants and propose adaptive ensemble strategies to enhance the generalization performance. PromptKD [50] adapts a larger CLIP teacher to downstream datasets and distills the knowledge to a smaller student in an unsupervised manner, separating the need for labeled domain data during transfer. TaskRes [75] proposes to decouple the prior knowledge of the pre-trained models and the task-specific knowledge, enabling reliable old knowledge preservation and flexible new knowledge exploration. GraphAdapter [48] further utilizes the dual-modality structure knowledge for better adaptation in downstream tasks.

**Agent collaboration.** Considering the complementary knowledge of diverse pre-trained models specialized in different domains or tasks, several works have been proposed to solve vision tasks with agent collaboration [74, 82, 72, 52, 71, 28]. The most relevant to our work is CaFo [81], which transfers the external knowledge using cache models [80]. However, such an ensemble manner introduces further costs in the inference stage. On the contrary, we adopt heterogeneous agent collaboration to aggregate the knowledge, and distillation strategy to inject the knowledge into CLIP, which demonstrates better performance and guarantees deployment efficiency.

**Multi-teacher distillation.** To improve the effectiveness of knowledge distillation [36], recent works [10, 30, 26, 76–78, 83, 54, 63] attempt to integrate the knowledge from multiple teacher networks. To be noted, how to perform multi-teacher distillation is not trivial, and how to extract and collaborate knowledge from various heterogeneous teachers has not been fully explored for CLIP-like foundation models. In this work, we devise a generic knowledge extraction method and flexible knowledge collaboration mechanism to further enhance the generalization ability of VL foundation models.

## 3 Method

In this section, we introduce our TransAgent in detail. As shown in Figure 1, it consists of vision-language foundation models and agents from different modalities. In the following section, we will illustrate how to collaborate with them for downstream visual recognition tasks. To start with, we briefly review the V-L foundation models by using the well-known CLIP [61]. Specifically, CLIP consists of two branches. In the vision branch, an image is first divided into $L$ non-overlapping equal-sized patches and then projected as input visual tokens $\mathbf{V}_{in}$, which are fed to the vision encoder to obtain image feature. In the language branch, a text description is projected as input textual tokens $\mathbf{T}_{in}$ which are then processed by the text encoder to generate text feature. Through contrastive learning over massive image-text pairs, CLIP achieves good alignment between the two modalities.

More interestingly, such large-scale V-L models show the potential in visual classification on the downstream tasks. By converting each class label into a text template such as "a photo of a {class

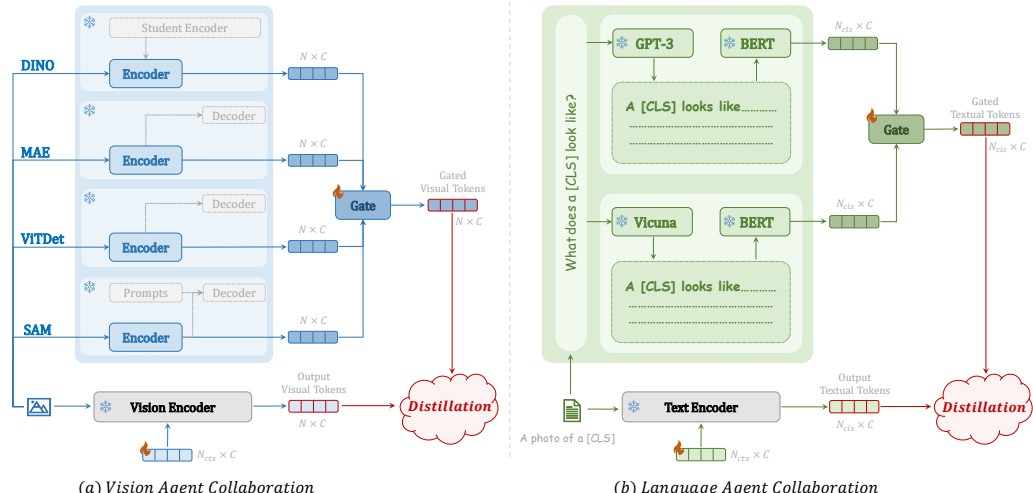

(a) Vision Agent Collaboration  (b) Language Agent Collaboration

Figure 2: **Vision Agent Collaboration and Language Agent Collaboration.** (a) VAC integrates visual knowledge via MoA gating and transfers the knowledge through layer-wise feature distillation. (b) LAC enhances the textual representations through class-specific feature distillation between the prompted textual feature and the gated textual feature.

name}", these models can easily achieve zero-shot inference. To further enhance their generalization ability under the few-shot settings, prompt learning methods are proposed, which introduce a number of learnable prompts while freezing the pre-trained CLIP [87, 86, 39–41, 45]. Following previous works, we add a set of learnable textual prompts $\mathbf{P}_T \in \mathbb{R}^{N_{ctx} \times C}$, where $N_{ctx}$ is the number of learnable prompts, in the language branch and concatenate them with the textual tokens, which are then processed by the text encoder to obtain the prompted textual feature $\mathbf{T} \in \mathbb{R}^{N_{cls} \times C}$ for all $N_{cls}$ categories. Similarly, a set of learnable visual prompts $\mathbf{P}_V$ are inserted in the image branch to generate the prompted visual feature $\mathbf{V} \in \mathbb{R}^{N \times C}$ where $N$ denotes the number of images:

$$\mathbf{T} = \text{TextEncoder}(\mathbf{T}_{in}, \mathbf{P}_T), \qquad \mathbf{V} = \text{VisionEncoder}(\mathbf{V}_{in}, \mathbf{P}_V). \tag{1}$$

Consequently, we compute the prediction score vectors $\mathbf{S} = \{\mathbf{S}^c\}$ for the image samples, where $\mathbf{S}^c$ is the cosine similarity between the visual feature $\mathbf{V}$ and the textual feature $\mathbf{T}^c$ of specific class $c$. As a result, we minimize the cross entropy loss between the score vectors and the ground truth labels:

$$\mathcal{L}_{\text{CE}} = \text{CrossEntropy}(\text{softmax}(\mathbf{S}), \mathbf{Y}), \tag{2}$$

to fine-tune the learnable prompts $\mathbf{P}_T$ and $\mathbf{P}_V$ for visual recognition. However, as mentioned in the introduction, it is difficult to achieve good generalization by adapting such a single CLIP model, especially when the domain shift of the target datasets is large. Hence, we propose to transfer diversified knowledge from heterogeneous agents in different modalities for better adaption of CLIP.

### 3.1 Vision Agent Collaboration (VAC)

One important component of CLIP is its vision branch in Figure 2. Hence, we consider to enhance this branch by transferring visual knowledge from various vision agents. To achieve this goal, we have to answer three critical questions. The first question is which models should be used for collaboration. As we know, CLIP mainly establishes image-level alignment between the two modalities, neglecting visual details in the pixel space. To fill this gap, we choose vision agents from two aspects. On one aspect, we choose vision models pre-trained with self-supervision such as MAE [33] and DINO [9]. Both agents focus on detailed image modeling via image masking [33] or patch self-distillation [9]. On the other aspect, we choose vision models built on dense supervision such as ViTDet [49] and SAM [43]. Both agents work on instance-level prediction with bounding boxes and masks. Detailed information on these models can be found in the supplementary.

The second question is how to extract visual knowledge by collaborating with these agents. As mentioned in the introduction, the latent feature is a common knowledge form among these heterogeneous models. Hence, given an input image, we extract the intermediate visual features

$\{\mathbf{V}_A(i) \in \mathbb{R}^{N \times C}\}$ from the vision encoders of these agents. Moreover, the contribution of different agents may vary among different domains. To fully exploit these agents in a unified manner, we introduce a Mixture-of-Agents (MoA) gating mechanism to adaptively integrate $\{\mathbf{V}_A(i)\}$ as the visual knowledge. Specifically, we concatenate all the agent features along the channel dimension $C$, and feed them into an MLP network to generate the gating weight $\mathbf{W}_V$. Next, we obtain the gated visual features $\mathbf{V}_A$ by computing the weighted sum over $\{\mathbf{V}_A(i)\}$:

$$\mathbf{W}_V = \text{MLP}(\text{Concat}(\{\mathbf{V}_A(i)\})), \quad \mathbf{V}_A = \sum_i \mathbf{W}_V(i) \mathbf{V}_A(i). \tag{3}$$

The final question is how to transfer the visual knowledge to enhance CLIP's vision encoder. We adopt feature distillation [36] where we compute the L1 loss between the prompted visual features $\mathbf{V}$ in Eq. 1 and the gated visual features of the vision agents:

$$\mathcal{L}_{\text{VAC}} = |\mathbf{V} - \mathbf{V}_A|. \tag{4}$$

Since the original CLIP is frozen, the above loss term allows us to fine-tune the learnable visual prompts $\mathbf{P}_V$ in Eq. 1 and MLP gating network in Eq. 3, which enables us to adaptively transfer external visual knowledge to the visual prompts to empower generalization ability.

## 3.2  Language Agent Collaboration (LAC)

The other important component of CLIP is its language branch in Figure 2. Similar to the vision branch, we consider three critical questions to transfer the textual knowledge from various language agents. Recent studies [60, 42] have shown that it is coarse to use a simple template (*e.g.*, "a photo of a {class name}") to describe a certain category. Hence, we first interact with the popular chatbots such as GPT-3 [7] and Vicuna [18] to enrich the class descriptions using queries like "What does a {class name} look like?". After obtaining the detailed descriptions from these chatbots, we use a text encoder (*e.g.*, BERT [22]) to extract the text features $\{\mathbf{T}_A(j) \in \mathbb{R}^{N_{cls} \times C}\}$ of all descriptions. To adaptively integrate $\{\mathbf{T}_A(j)\}$ as the textual knowledge, we also utilize the MoA gating mechanism:

$$\mathbf{W}_T = \text{MLP}(\text{Concat}(\{\mathbf{T}_A(j)\})), \quad \mathbf{T}_A = \sum_j \mathbf{W}_T(j) \mathbf{T}_A(j). \tag{5}$$

Finally, for each category, we perform feature distillation where we compute the L1 loss between the prompted textual feature in Eq. 1 and the gated textual feature from language agents in Eq. 5:

$$\mathcal{L}_{\text{LAC}} = |\mathbf{T} - \mathbf{T}_A|. \tag{6}$$

By fine-tuning the learnable textual prompts $\mathbf{P}_T$ in Eq. 1 and MLP gating network in Eq. 5, we adaptively transfer the rich textual knowledge to enhance CLIP's textual representations.

## 3.3  Multi-modal Agent Collaboration (MAC)

Through vision and language agent collaboration, we can enhance the learnable visual and textual prompts respectively. Considering the key success in vision-language foundation models is credited to the multi-modal alignment, we investigate how to further align the visual and textual prompts with external multi-modal agents. First, there exist two types of multi-modal agents, including Text-to-Image (T2I) generative models [64, 11] and Image-to-Text (I2T) captioning models [46, 12]. Since both types of agents involve conversion from one modality to the other, we believe they implicitly achieve vision-language alignment. Based on the above discussion, we choose our T2I agents built upon two mainstream structures including Stable Diffusion [64] in UNet style and PixArt-$\alpha$ [11] in DiT style. Moreover, we choose our I2T agents specialized in two mainstream tasks including BLIP-2 [46] for general captioning and Shikra [12] for grounded dialogue.

Next, we consider what kind of knowledge can represent vision-language alignment. The most direct form may be the probability score vector that shows the prediction confidence over target classes. Hence, for each training image, we investigate such knowledge in these multi-modal agents.

In the T2I agents, cross attention effectively encodes relations between image and text [34, 84]. Hence, we leverage such prior to extract the score vector. Specifically, we use the template description of a class $c$ as text. Then, we extract cross attention value $\mathbf{M}_k^c$ between the text and the $k$-th token of the input image from the pre-trained T2I agents. Consequently, we sum over all the tokens to

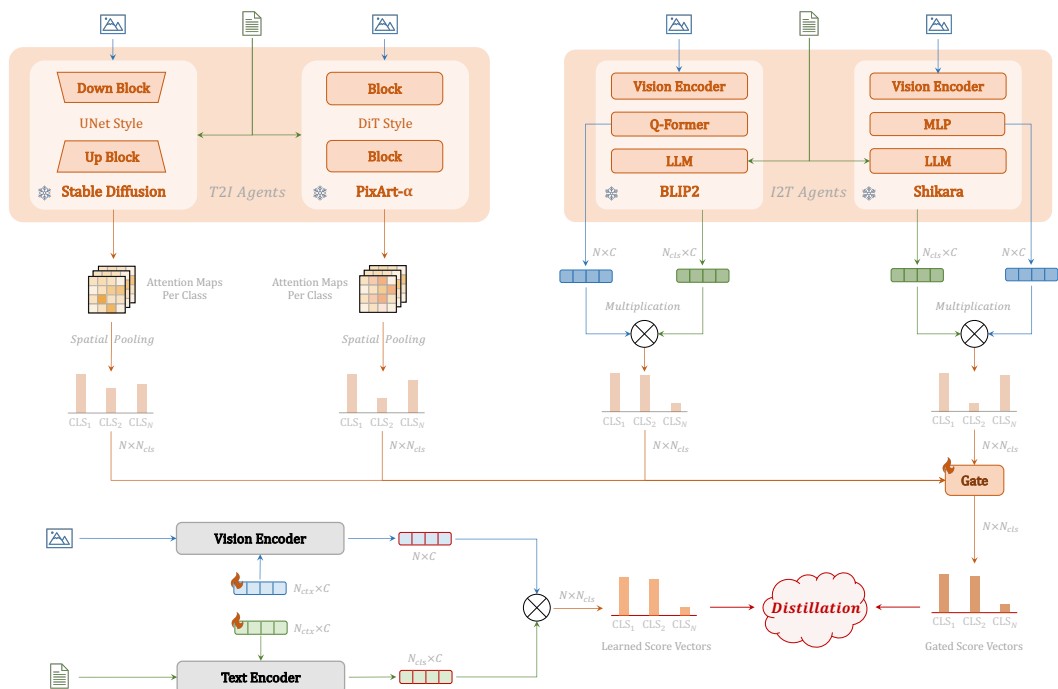

Figure 3: **Multi-modal Agent Collaboration.** *Top left:* We first extract the cross attention maps from the T2I agents and then obtain the score vectors through LSE pooling. *Top right:* We compute the score vectors from the I2T agents as the cosine similarity between the projected visual feature and the LLM's textual feature. Finally, we perform score distillation between the learned score vectors and the gated score vectors to further align the learnable prompts.

obtain the prediction score of the image *w.r.t.* class $c$: $\mathbf{S}_{T2I}^c = \log(\sum_k \exp(\mathbf{M}_k^c))$, where we adopt LogSumExp (LSE) pooling [5] to provide more accurate matching scores.

In the I2T agents, there exists a projection module (*e.g.*, Q-Former in BLIP-2 [46] and MLP in Shikra [12]) to adapt the visual features for the large language model. Hence, we extract the visual feature of an input image via the projection and the textual features of all the classes from the LLM. By computing the cosine similarity between them, we can obtain the prediction score $\mathbf{S}_{I2T}$ of the image over all the classes. Next, we leverage the MoA gating mechanism to adaptively summarize the prediction score vectors from all the multi-modal agents $\mathbf{M}_A = \text{Concat}(\{\mathbf{S}_{T2I}, \mathbf{S}_{I2T}\})$:

$$\mathbf{W}_S = \text{MLP}(\mathbf{M}_A), \qquad \mathbf{S}_A = \sum_n \mathbf{W}_S(n)\mathbf{M}_A(n). \tag{7}$$

Finally, we explore how to transfer $\mathbf{S}_A$ to enhance the learnable prompts in CLIP. Specifically, after being processed by the vision and text encoders of CLIP, the learnable visual and textual prompts $\mathbf{P}_V$ and $\mathbf{P}_T$ are transformed into $\mathbf{Q}_V$ and $\mathbf{Q}_T$. Unlike $\mathbf{P}_V$ and $\mathbf{P}_T$ which are universal, $\mathbf{Q}_V$ is relevant to the image samples and $\mathbf{Q}_T$ is relevant to the target classes. Hence, we can compute the cosine similarity between $\mathbf{Q}_V$ and $\mathbf{Q}_T$ to obtain the learned score vectors $\mathbf{S}_P \in \mathbb{R}^{N \times N_{cls}}$ of the input images over all the classes. We perform score distillation between $\mathbf{S}_P$ and $\mathbf{S}_A$:

$$\mathcal{L}_{\text{MAC}} = \text{KL}(\text{softmax}(\mathbf{S}_P)||\text{softmax}(\mathbf{S}_A)). \tag{8}$$

Through computing the KL divergence, we can leverage external multi-modal knowledge $\mathbf{S}_A$ as a semantic guidance to align the learnable visual and textual prompts.

### 3.4 Multi-Source Knowledge Distillation

Finally, we combine all the distillation loss from multiple sources, achieving heterogeneous agent collaboration for knowledge transfer:

$$\mathcal{L}_{\text{TransAgent}} = \mathcal{L}_{\text{CE}} + \lambda_1 \mathcal{L}_{\text{VAC}} + \lambda_2 \mathcal{L}_{\text{LAC}} + \lambda_3 \mathcal{L}_{\text{MAC}}, \tag{9}$$

Table 1: **Accuracy comparison with state-of-the-art methods on base-to-novel generalization.** All methods use CLIP's ViT-B/16 as the vision encoder. Our TransAgent exhibits strong generalization ability and outperforms previous SOTA on all datasets. The best results are **bolded**.

| Method | Average | | | ImageNet [21] | | | Caltech101 [29] | | | OxfordPets [58] | | |
|---|---|---|---|---|---|---|---|---|---|---|---|---|
| | Base | Novel | HM | Base | Novel | HM | Base | Novel | HM | Base | Novel | HM |
| CLIP [61] | 69.34 | 74.22 | 71.70 | 72.43 | 68.14 | 70.22 | 96.84 | 94.00 | 95.40 | 91.17 | 97.26 | 94.12 |
| CoOp [87] | 82.69 | 63.22 | 71.66 | 76.47 | 67.88 | 71.92 | 98.00 | 89.81 | 93.73 | 93.67 | 95.29 | 94.47 |
| CoCoOp [86] | 80.47 | 71.69 | 75.83 | 75.98 | 70.43 | 73.10 | 97.96 | 93.81 | 95.84 | 95.20 | 97.69 | 96.43 |
| MaPLe [40] | 82.28 | 75.14 | 78.55 | 75.40 | 70.32 | 72.72 | 98.27 | 93.23 | 95.68 | 95.43 | 97.83 | 96.62 |
| RPO [45] | 81.13 | 75.00 | 77.78 | 76.60 | **71.57** | 74.00 | 97.97 | 94.37 | 96.03 | 94.63 | 97.50 | 96.05 |
| PromptSRC [41] | 84.26 | 76.10 | 79.97 | 77.60 | 70.73 | 74.01 | 98.10 | 94.03 | 96.02 | 95.33 | 97.30 | 96.30 |
| **TransAgent** | **85.29** | **77.62** | **81.27** | **78.07** | 70.57 | **74.13** | **98.90** | **95.23** | **97.03** | **96.33** | **98.13** | **97.22** |

| Method | StanfordCars [44] | | | Flowers102 [57] | | | Food101 [6] | | | FGVCAircraft [55] | | |
|---|---|---|---|---|---|---|---|---|---|---|---|---|
| | Base | Novel | HM | Base | Novel | HM | Base | Novel | HM | Base | Novel | HM |
| CLIP [61] | 63.37 | 74.89 | 68.65 | 72.08 | 77.80 | 74.83 | 90.10 | 91.22 | 90.66 | 27.19 | 36.29 | 31.09 |
| CoOp [87] | 78.12 | 60.40 | 68.13 | 97.60 | 59.67 | 74.06 | 88.33 | 82.26 | 85.19 | 40.44 | 22.30 | 28.75 |
| CoCoOp [86] | 70.49 | 73.59 | 72.01 | 94.87 | 71.75 | 81.71 | 90.70 | 91.29 | 90.99 | 33.41 | 23.71 | 27.74 |
| MaPLe [40] | 74.70 | 71.20 | 72.91 | 97.70 | 68.68 | 80.66 | 90.30 | 88.57 | 89.43 | 36.90 | 34.13 | 35.46 |
| RPO [45] | 73.87 | **75.53** | 74.69 | 94.13 | 76.67 | 84.50 | 90.33 | 90.83 | 90.58 | 37.33 | 34.20 | 35.70 |
| PromptSRC [41] | 78.27 | 74.97 | 76.58 | 98.07 | 76.50 | 85.95 | 90.67 | 91.53 | 91.10 | 42.73 | 37.87 | 40.15 |
| **TransAgent** | **79.53** | 74.73 | **77.06** | **98.37** | **77.13** | **86.46** | **90.87** | **92.20** | **91.53** | **43.77** | **39.00** | **41.25** |

| Method | SUN397 [73] | | | DTD [19] | | | EuroSAT [35] | | | UCF101 [67] | | |
|---|---|---|---|---|---|---|---|---|---|---|---|---|
| | Base | Novel | HM | Base | Novel | HM | Base | Novel | HM | Base | Novel | HM |
| CLIP [61] | 69.36 | 75.35 | 72.23 | 53.24 | 59.90 | 56.37 | 56.48 | 64.05 | 60.03 | 70.53 | 77.50 | 73.85 |
| CoOp [87] | 80.60 | 65.89 | 72.51 | 79.44 | 41.18 | 54.24 | 92.19 | 54.74 | 68.69 | 84.69 | 56.05 | 67.46 |
| CoCoOp [86] | 79.74 | 76.86 | 78.27 | 77.01 | 56.00 | 64.85 | 87.49 | 60.04 | 71.21 | 82.33 | 73.45 | 77.64 |
| MaPLe [40] | 78.47 | 76.93 | 77.79 | 80.67 | 56.48 | 66.44 | 83.90 | 66.00 | 73.88 | 85.23 | 71.97 | 78.04 |
| RPO [45] | 80.60 | 77.80 | 79.18 | 76.70 | 62.13 | 68.61 | 86.63 | 68.97 | 76.79 | 83.67 | 75.43 | 79.34 |
| PromptSRC [41] | 82.67 | 78.47 | 80.52 | 83.37 | 62.97 | 71.75 | 92.90 | 73.90 | 82.32 | 87.10 | 78.80 | 82.74 |
| **TransAgent** | **82.90** | **79.30** | **81.06** | **84.37** | **63.67** | **72.57** | **97.43** | **83.43** | **89.89** | **87.60** | **80.47** | **83.88** |

where $\lambda_1$, $\lambda_2$, $\lambda_3$ are hyperparameters. Since all the pre-trained models are frozen in the training phase, we only need to fine-tune the learnable vision and language prompts with negligible cost. More importantly, owing to the distillation strategy, all the agents can be unloaded and the modality-specific gates can be abandoned in the inference phase. We can simply use the enhanced CLIP just like the original one, which largely boosts deployment efficiency without a model ensemble.

## 4 Experiments

**Datasets and Metrics.** We evaluate our proposed method on 11 commonly used datasets covering a wide range of recognition tasks, including ImageNet [21], Caltech101 [29], OxfordPets [58], StanfordCars [44], Flowers102 [57], Food101 [6], FGVCAircraft [55], SUN397 [73], UCF101 [67], DTD [19] and EuroSAT [35]. We explore two typical low-shot scenarios to evaluate the performance. (i) Base-to-novel generalization: The datasets are equally split into base and novel classes. The model is trained on base classes and evaluated on the test set of both classes. We report the base and novel class accuracy and the harmonic mean (HM) of the results. (ii) Few-shot classification: We assess the accuracy trained with 1/2/4/8/16 shot(s) per class to examine the model's learning capacity.

**Implementation Details.** We adopt CLIP ViT/B-16 as our backbone and conduct all experiments using 3 different seeds to obtain an averaged result, following previous works [40, 41, 50]. Our method ensembles knowledge from heterogeneous agents, including pre-trained vision models, LLMs, T2I generative models, and I2T captioning models. Detailed information on the agents and training settings are shown in Appendix A.

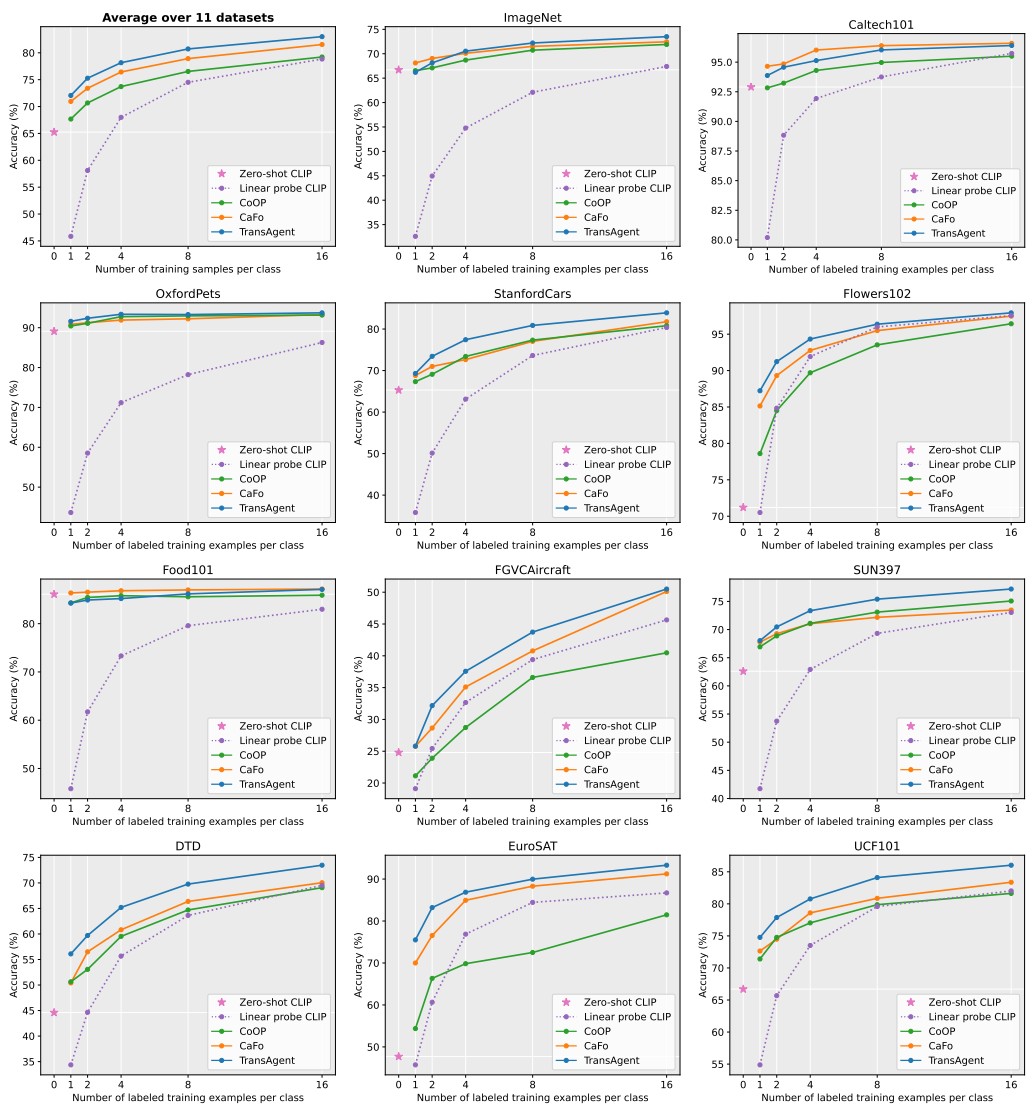

Figure 4: **Accuracy comparison in few-shot classification.** TransAgent demonstrates *state-of-the-art* performance for all few-shot settings on different datasets, which proves promising learning capability even under extremely limited supervision.

## 4.1 Comparison with State-of-the-Art

In Table 1, we compare the base-to-novel generalization performance of our proposed TransAgent with state-of-the-art prompt learning methods, including CoOp [87], CoCoOp [86], MaPLe [40], RPO [45] and PromptSRC [41]. All approaches use the same CLIP ViT-B/16 during the evaluation stage. Our method demonstrates superior performance across 11 datasets, surpassing all competing methods in terms of base and novel accuracy, as well as the harmonic mean, particularly excelling on the EuroSAT dataset. In Figure 4, we present the few-shot classification performance of TransAgent and previous methods. Our method consistently outperforms the counterparts and demonstrates growing learning capability when the training samples increase. To be mentioned, TransAgent outperforms CaFo [81] with much lower deployment costs (see Table 13). Detailed comparisons on cross-dataset evaluation and domain generalization are provided in Appendix B.

## 4.2 Ablative Analysis

In this section, we present the ablative analysis of our agent collaboration designs for each modality. We adopt IVLP [40, 41] as our baseline model.

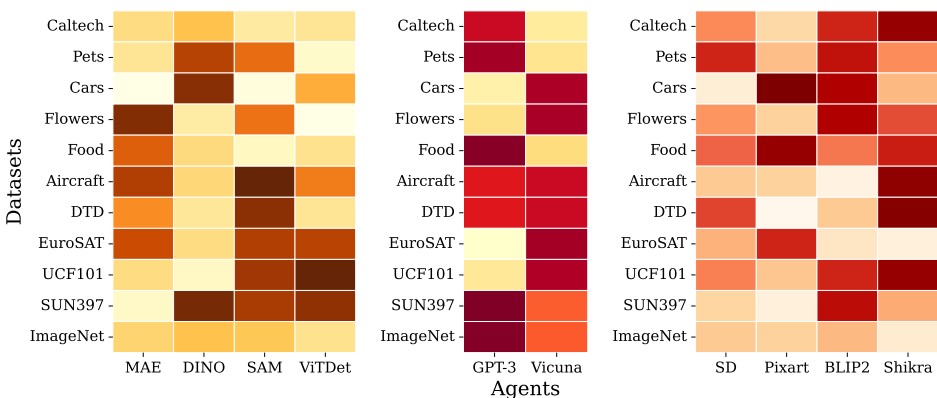

Figure 5: **Averaged gating weights of each agent on different datasets.** Deeper color indicates more contributions to the gated feature(s) or score vectors.

Table 2: **VAC Design.**

| Models | Base | Novel | HM |
|---|---|---|---|
| baseline | 84.21 | 71.79 | 77.51 |
| MAE | 84.51 | 71.66 | 77.55 |
| DINO | 84.60 | 70.90 | 77.15 |
| SAM | 84.73 | 71.87 | 77.77 |
| ViTDet | 84.45 | 72.04 | 77.75 |
| Last-layer | 84.47 | 75.92 | 79.97 |
| Layer-wise | 85.29 | 77.62 | 81.27 |
| Average | 84.20 | 74.79 | 79.22 |
| Add | 84.40 | 75.17 | 79.52 |
| Gating | 85.29 | 77.62 | 81.27 |

Table 3: **LAC Design.**

| Models | Base | Novel | HM |
|---|---|---|---|
| baseline | 84.21 | 71.79 | 77.51 |
| GPT-3 | 85.15 | 74.55 | 79.50 |
| Vicuna | 85.35 | 74.70 | 79.67 |
| $[SOS]$ | 84.19 | 75.25 | 79.47 |
| $[EOS]$ | 85.29 | 77.62 | 81.27 |
| Average | 84.23 | 75.98 | 79.89 |
| Add | 82.44 | 74.89 | 78.48 |
| Gating | 85.29 | 77.62 | 81.27 |

Table 4: **MAC Design.**

| Models | Base | Novel | HM |
|---|---|---|---|
| baseline | 84.21 | 71.79 | 77.51 |
| Stable Diffusion | 84.91 | 73.11 | 78.57 |
| Pixart-$\alpha$ | 84.78 | 73.47 | 78.72 |
| BLIP2 | 84.87 | 73.43 | 78.73 |
| Shikra | 84.97 | 72.90 | 78.47 |
| Prompted logits | 84.45 | 75.18 | 79.54 |
| Learned scores | 85.29 | 77.62 | 81.27 |
| Average | 84.33 | 75.04 | 79.41 |
| Add | 84.37 | 75.35 | 79.61 |
| Gating | 85.29 | 77.62 | 81.27 |

Table 5: **Performance w/ individual module.**

| VAC | LAC | MAC | HM | Δ |
|---|---|---|---|---|
| ✗ | ✗ | ✗ | 77.51 | - |
| ✓ | ✗ | ✗ | 79.04 | +1.53 |
| ✗ | ✓ | ✗ | 79.90 | +2.39 |
| ✗ | ✗ | ✓ | 79.61 | +2.10 |

Table 6: **Performance w/ module combinations.**

| VAC | LAC | MAC | HM | Δ |
|---|---|---|---|---|
| ✓ | ✓ | ✗ | 80.02 | +2.51 |
| ✓ | ✗ | ✓ | 79.79 | +2.28 |
| ✗ | ✓ | ✓ | 80.40 | +2.89 |
| ✓ | ✓ | ✓ | 81.27 | +3.76 |

**Effectiveness of individual agent.** In Table 2 - Table 4, we show the results of introducing an individual agent as a teacher to supervise the baseline model in the beginning rows. Nearly all vision agents contribute to the improvement in accuracy, except for DINO which is behind. However, we observe that DINO performs well on most datasets, except on EuroSAT (with a 5.60% decline). Nonetheless, we still keep DINO as one of the vision agents. While for the language and the multi-modal agents, they all boost the performance of the baseline model respectively.

**Distillation strategy.** In the middle rows of Table 2 - Table 4, we ablate the alignment strategy for agent collaboration of each modality. (i) For VAC, we choose to adopt feature distillation with either the average-pooled feature at the last layer or all features at all layers. As the results suggest, layer-wise distillation of all features performs better. (ii) For LAC, two special tokens from the output of the text encoder are considered to compute Equation 6. These tokens are inserted into the text sequence to compose a complete prompt. $[EOS]$ performs better owing to the causal attention module in Transformer blocks, so that it aggregates more information. (iii) For MAC, either the prompted logits of CLIP or the learned scores (Eq. 8) are chosen to align with the semantic logits from multi-modal agents. Since the prompted logits need to be aligned with ground-truth labels as well (Eq. 2), there might be confusion to align it with the external knowledge simultaneously. Experimental results show that using learned scores to align with semantic logits yields better results.

**Fusion design.** In the last rows of Table 2 - Table 4, we ablate the fusion design for each modality. "Average" refers to simply calculating the average of all output features from the agents along the channel dimension; "Add" refers to calculating the distillation loss separately for each agent; "Gating" denotes our proposed MoA gating mechanism. As can be seen, gating fusion achieves the best results for all collaboration designs, since it adaptively selects the useful information from the agents.

**Effectiveness of collaboration module(s).** As presented in Table 5, the first row indicates the baseline model. Each individual module is beneficial for improving the generalization ability of the foundation models, where the last column shows the relative increase. Results in Table 6 demonstrate that the combinations of collaboration modules further boost the results.

### 4.3 Visualization

We calculate the averaged gating weights of each agent with 16-shot training samples from all classes of a certain dataset and present the results with heat maps. As shown in Figure 5, different agents do not contribute equally towards certain target datasets, which also verifies the superiority of our gating fusion design. For vision agents, DINO experts in recognizing general objects while lags behind on some fine-grained datasets (*e.g.*, EuroSAT), whereas the others that focus more on details perform better. Language agents provide domain-specific linguistic knowledge accordingly. I2T agents consistently provide their knowledge thanks to the grounding nature, while T2I agents demonstrate better capability in fine-grained scenarios.

## 5 Conclusion

We propose TransAgent, a unified framework to transfer vision-language foundation models through heterogeneous agent collaboration. By adaptively integrating the external knowledge of agents from different modalities viac the MoA gating mechanism, TransAgent achieves *state-of-the-art* performance on 11 datasets under the low-shot scenarios.

**Limitations.** Although TransAgent collaborates with heterogeneous agents in a unified manner, transferring the external knowledge through distillation may harm the original CLIP's representations even using prompt learning methods because the domain knowledge from agents are diversified, and irrelevant information may also be introduced. Moreover, one of the key characteristics of these agents is large-scale pre-training, few-shot scenarios may not meet their data-hungry nature. So our future work would be integrating the knowledge with directed focus and further unleashing the potential of the agents even when the domain-specific knowledge (*e.g.*, labels) is absent.

## Acknowledgments and Disclosure of Funding

This work was supported in part by the National Key R&D Program of China (NO.2022ZD0160505), the National Natural Science Foundation of China under Grant (62272450)

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

Table 7: **Demonstration of heterogeneous agents specialized in different domains or tasks.**

| Model | Parameters | Model Type | Pre-training | | Knowledge |
|---|---|---|---|---|---|
| | | | Tasks | Datasets | |
| DINO [9] | 86M | ViT | IC | ImageNet-1K | Vision |
| MAE [33] | 86M | ViT | MIM | ImageNet-1K | Vision |
| SAM [43] | 86M | ViT | IS | SA-1B | Vision |
| ViTDet [49] | 86M | ViT | OD | COCO | Vision |
| GPT-3 [7] | 175B | LLM | TG | - | Language |
| Vicuna [18] | 13B | LLM | TG | - | Language |
| BERT [22] | 38M | Transformer | MLM | - | Language |
| Stable Diffusion [64] | 0.86B | UNet | IG | LAION-2B | Multi-modal |
| Pixart-$\alpha$ [11] | 0.6B | DiT | IG | - | Multi-modal |
| BLIP-2 [46] | 2.7B | MLLM | ITC+ITM+ITG | - | Multi-modal |
| Shikra [12] | 7B | MLLM | ITC+ITM+ITG | - | Multi-modal |

Table 8: **Memory and training time required for each dataset.**

| | Setting | ImageNet | Caltech 101 | Oxford Pets | Standford Cars | Flowers 102 | Food101 | FGVC Aircraft | SUN397 | DTD | Euro SAT | UCF101 |
|---|---|---|---|---|---|---|---|---|---|---|---|---|
| **Memory (MB)** | Base-to-novel | 11790 | 4708 | 4692 | 6698 | 4702 | 4708 | 5286 | 7708 | 5008 | 4034 | 4718 |
| | Few-shot (16-shot) | 19978 | 4542 | 6892 | 6892 | 5488 | 5488 | 5486 | 10110 | 4700 | 4124 | 5484 |
| **Time (ms/batch)** | Base-to-novel | 400 | 190 | 195 | 226 | 211 | 209 | 219 | 273 | 191 | 190 | 193 |
| | Few-shot (16-shot) | 688 | 214 | 206 | 262 | 217 | 215 | 221 | 356 | 196 | 205 | 220 |

# A   Additional Implementation Details

**Training Details.**   The number of learnable vision and language prompt tokens are both set to 4, and the prompt depth is set to 9 for base-to-novel generalization and few-shot classification, and 3 for cross-dataset and domain generalization. The learnable text prompts of the first layer are initialized with the word embeddings of "a photo of a", while the other learnable prompts are randomly initialized with a normal distribution. For vision agent collaboration, we adopt online distillation where the models are loaded during the training process. For language and multi-modal agent collaboration, due to the huge model parameters, we extract their knowledge offline before our training launches. For few-shot classification, we train the models for 50 epochs under different low-shot settings (ranging from 1 to 16). We re-implement the compared methods in Figure 4 with Vision Transformer [24] backbone for fair comparison. For the other benchmarks, we observe that PromptKD [50] adopts a transductive setting which uses all training samples in an unsupervised manner and obtains better results. However, such a setting is beyond our scope. In our work, all models are trained for 20 epochs using 16-shot samples with a fixed batch size of 4 and a learning rate of 0.0025 with SGD as the optimizer. We set $\lambda_1 = 1$, $\lambda_2 = 25$ and $\lambda_3 = 1$ in Eq. 9 after extensive hyperparameter search to balance the total loss. The memory and training time needed under different settings on each dataset are provided in Table 8 using a batch size of 4. At least 48GB is required to extract knowledge from language and multi-modal agents. All experiments are conducted on a single Nvidia A6000 GPU.

**Information of agents.**   Table 7 lists all the agents we have collaborated in our TransAgent framework. (i) Vision agents: DINO provides robust representations through image contrastive (IC) learning while MAE presents powerful capability via mask image modeling (MIM). SAM and ViT-Det are specialists in image segmentation (IS) and object detection (OD) respectively. (ii) Language agents: Both GPT-3 and Vicuna are excellent chatbots that faithfully follow human instructions and respond with convincing answers by text generation (TG). We use BERT, which is just CLIP's text encoder pre-trained using mask language modeling (MLM), to process the generated captions from chatbots before transferring these linguistic knowledge. (iii) Multi-modal agents: Stable Diffusion and Pixart-$\alpha$ share different model architectures, but both T2I agents demonstrate astonishing image generation (IG) capability. The I2T agents are pre-trained with multiple tasks including image-text contrastive (ITC), image-text matching (ITM), and image-grounded text generation (ITG) over large-scale datasets. These agents demonstrate outstanding multi-modal understanding ability.

Table 9: **Accuracy comparison with previous methods on cross-dataset evaluation.**

| Method | Source | Target | | | | | | | | | | |
|---|---|---|---|---|---|---|---|---|---|---|---|---|
| | ImageNet | Caltech 101 | Oxford Pets | Standford Cars | Flowers 102 | Food101 | FGVC Aircraft | SUN397 | DTD | Euro SAT | UCF101 | Avg. |
| CLIP | 66.72 | 92.94 | 89.07 | 65.29 | 71.30 | 86.11 | **24.87** | 62.62 | 44.56 | 47.69 | 66.77 | 65.12 |
| CoOp | 71.51 | 93.70 | 89.14 | 64.51 | 68.71 | 85.30 | 18.47 | 64.15 | 41.92 | 46.39 | 66.55 | 63.88 |
| CoCoOp | 71.02 | **94.43** | 90.14 | 65.32 | 71.88 | 86.06 | 22.94 | **67.36** | 45.73 | 45.37 | 68.21 | 65.74 |
| MaPLe | 70.72 | 93.53 | **90.49** | 65.57 | **72.23** | 86.20 | 24.74 | 67.01 | 46.49 | 48.06 | 68.69 | 66.30 |
| PromptSRC | 71.27 | 93.60 | 90.25 | **65.70** | 70.25 | 86.15 | 23.90 | 67.10 | **46.87** | 45.50 | 68.75 | 65.81 |
| **TransAgent** | **72.00** | 94.37 | 90.33 | 65.43 | 71.40 | **86.47** | 23.20 | 66.20 | 45.30 | **52.13** | **69.93** | **66.48** |

Table 10: **Accuracy comparison on domain generalization.**

| Method | Source | Target | | | | |
|---|---|---|---|---|---|---|
| | ImageNet | -V2 | -S | -A | -R | Avg. |
| CLIP | 66.73 | 60.83 | 46.15 | 47.77 | 73.96 | 57.18 |
| CoOp | 71.51 | 64.20 | 47.99 | 49.71 | 75.21 | 59.28 |
| CoCoOp | 71.02 | 64.07 | 48.75 | 50.63 | 76.18 | 59.91 |
| MaPLe | 70.72 | 64.07 | 49.15 | 50.90 | 76.98 | 60.27 |
| PromptSRC | 71.27 | 64.35 | 49.55 | 50.90 | **77.80** | 60.65 |
| **TransAgent** | **72.00** | **64.87** | **49.63** | **51.23** | 77.53 | **60.82** |

Table 11: **MAC loss type.**

| Loss | Base | Novel | HM |
|---|---|---|---|
| L1 | 84.93 | 74.69 | 79.48 |
| MSE | 84.91 | 74.40 | 79.31 |
| **KL** | **85.29** | **77.62** | **81.27** |

Table 12: **Pooling type.**

| Pooling | Base | Novel | HM |
|---|---|---|---|
| Average | 84.97 | 76.12 | 80.30 |
| Max | 84.69 | 75.12 | 79.62 |
| **LogSumExp** | **85.29** | **77.62** | **81.27** |

Table 13: **Inference time.**

| Method | Time | Accuracy |
|---|---|---|
| Zero-shot CLIP | **1min06s** | 66.70 |
| CaFo | 16min14s | 72.46 |
| **TransAgent** | 2min35s | **73.50** |

# B   Additional Experiments

**Cross-Dataset Evaluation**. We compare the cross-dataset performance in Table 9. On the source dataset, TransAgent achieves better performance than previous few-shot prompt learning methods under the same training settings. We credit the fine-fitted result to the ImageNet pre-trained vision experts, which transfer their knowledge of the source dataset during the training process. More surprisingly, our method does not seem to overfit on the source dataset. It still presents competitive results against MaPLe, which is the SOTA few-shot method on cross-dataset generalization, leading to an overall improvement over the previous methods.

**Domain Generalization**. Table 10 presents the results on domain generalization. TransAgent outperforms previous few-shot prompt learning methods with the highest average accuracy. This suggests that our method improves the robustness of VLMs against out-of-distribution data.

# C   Additional Ablative Analysis

**MAC loss type.** We scrutinize the loss function used to compute the distillation loss in Eq. 8. As shown in Table 11, using L1 loss or MSE loss underperforms using KL loss by a large margin. Such channel-wise loss is incompatible with the form of the score vectors, which are per-class probability distributions, and using such loss makes it hard to converge. On the contrary, KL loss provides smoother training due to the soft matching between the probability distributions (*i.e.*, the score vectors).

**Pooling type.** In Table 12, we ablate the pooling strategy to aggregate the cross attention value from T2I agents to obtain the score vectors. The pooling operations are performed on the spatial dimension of the cross attention maps, which results in the semantic logits of a certain image over all the candidate categories. As is shown in Table 12, average and max pooling lag behind since they introduce information loss during the operations. However, LogSumExp (LSE) pooling yields the best performance as it is more robust and provides more accurate matching scores [34].

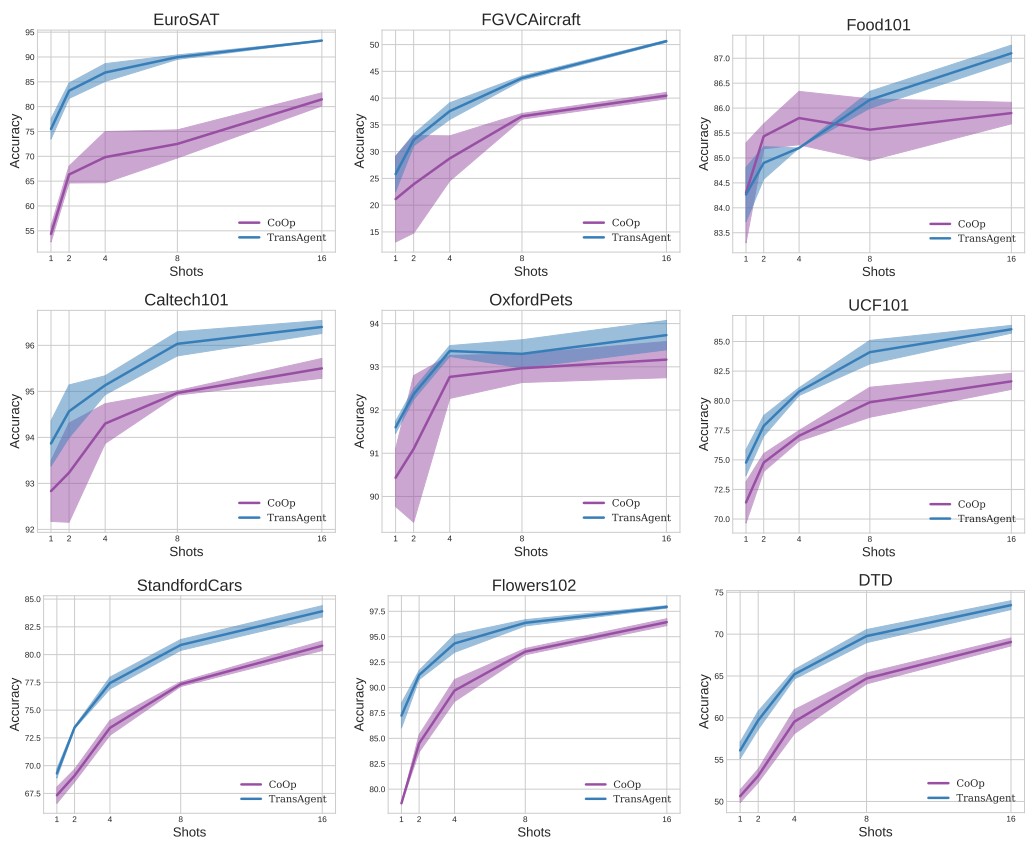

Figure 6: **Variance and performance of TransAgent compared with CoOp.** TransAgent demonstrates better robustness and outperforms CoOp on most low-shot cases.

**Deployment efficiency.** Table 13 presents the comparison of inference time and performance evaluated on the test set of ImageNet [21]. All models adopt Vision Transformer [24] backbone. Our method demonstrates superior deployment efficiency compared to CaFo [81] which adopts a cumbersome model ensemble. To be mentioned, the learnable prompts in the training phase are frozen during the inference stage, which carries the knowledge of heterogeneous agents to enhance CLIP's generalization ability. The inference time of our TransAgent is a little behind the zero-shot CLIP, but it still achieves a good accuracy-efficiency tradeoff.

## D    Additional Visualization

To verify the robustness of TransAgent, we run few-shot classification experiments over 3 seeds and visualize the variance on 9 datasets in Figure 6 (ImageNet and SUN397 are excluded since the variance is small for both methods.). We can observe that TransAgent consistently outperforms CoOp in robustness and excels in most low-shot scenarios.

## E    Examples of Generated Descriptions from Chatbots

We interact with the language agents (*i.e.*, chatbots) as described in Section 3.2 to obtain rich descriptions of the categories in the target domain, following [42]. To further unleash the linguistic potential of these language agents, we gather their knowledge adaptively via the MoA gating mechanism. In fact, these language agents give descriptions in different styles with their respective unique understanding. Below we show some examples of the descriptions given by different chatbots for certain classes to demonstrate their characteristics.

**Class: Pomeranian (OxfordPets).**

GPT-3: "A pet Pomeranian typically has a thick coat of fur that can be either straight or curly."

Vicuna: "The image shows a small dog with a fluffy white coat. The dog has a round face with big brown eyes and a black nose. Its ears are small and pointy, and its tail is curled up over its back."

**Class: Industrial Buildings (EuroSAT).**

GPT-3: "A satellite photo of industrial buildings might look like a patchwork of utilitarian structures."

Vicuna: "The image shows a large industrial building with a flat roof and several windows on the front facade. The building is made of concrete and has a gray color. The roof is also gray and appears to be made of a similar material."

**Class: Club Sandwich (Food101).**

GPT-3: "A club sandwich is a sandwich made with ham, turkey, bacon, cheese, and tomato, with lettuce in between slices of bread."

Vicuna: "The image shows a golden brown club sandwich, with a thick slice of bread on the bottom, and two thinner slices on top. The bread is toasted and has a crispy texture."

**Class: Office Building (SUN397).**

GPT-3: "An office building is usually a tall building with many floors."

Vicuna: "An office building is a structure that is designed for business-related activities. It typically has multiple floors and a variety of rooms and spaces to accommodate different functions, such as open-plan offices, meeting rooms, and private offices."

**Class: 747-200 (FGVCAircraft).**

GPT-3: "The 747-200 is a jumbo jet produced by Boeing."

Vicuna: "The Boeing 747-200 is a large, wide-body commercial airliner that was first introduced in the 1970s."

**Class: 2012 Mercedes-Benz E-Class Sedan (StandfordCars).**

GPT-3: "The Mercedes-Benz E-Class Sedan 2012 can be identified by its distinctive grille, sleek headlights, and classic Mercedes-Benz logo."

Vicuna: "The 2012 Mercedes-Benz E-Class Sedan is a luxury car that is known for its sleek and sophisticated appearance."

