# OpenReview forum: "TransAgent: Transfer Vision-Language Foundation Models with Heterogeneous Agent Collaboration"
_NeurIPS.cc/2024/Conference — NeurIPS 2024 poster_

### Official Review · Reviewer_DUuV · 2024-07-05

**Soundness:** 2
**Presentation:** 3
**Contribution:** 3
**Rating:** 6
**Confidence:** 5

**Summary:**

This paper proposes the TransAgent framework, which unifies and transports knowledge from isolated agents to guide CLIP in generalizing through multi-source knowledge distillation. This framework allows flexible collaboration with 11 heterogeneous agents to enhance vision-language foundation models. Importantly, this approach incurs no additional cost during the inference phase.

**Strengths:**

**[Interesting idea]** Leveraging heterogeneous agent collaboration for generalized vision language models is intriguing.

**[Good presentation]** The paper is well-written and organized, which is easy to follow.

**[Well-illustrated figures]** The figures shown in this paper are clear enough to tell the workflow of the method.

**Weaknesses:**

**[Overclaimed statements]** The paper mentions that “it is the first unified transfer framework for generalizing vision-language foundation models with heterogeneous agent collaboration.” However, CaFo [67] had done similar things by adopting multiple heterogeneous agents. This statement may need a revision to differentiate from CaFo.

**[Some SOTA methods are not compared]** Some works to be compared or discussed in related works are listed as follow: “PromptKD: Unsupervised Prompt Distillation for Vision-Language Models, CVPR2024”, “DePT: Decoupled Prompt Tuning, CVPR2024”, “Beyond Sole Strength: Customized Ensembles for Generalized Vision-Language Models, ICML2024”, “Consistency-guided Prompt Learning for Vision-Language Models, ICLR2024”, “GraphAdapter: Tuning Vision-Language Models with Dual Knowledge Graph, NeurIPS2023” and “Task Residual for Tuning Vision-Language Models, CVPR2023”.

**[Need further explorations]** The knowledge distillation technique used in this work is too straightforward. It would be nice to see an advanced distillation with heterogeneous agents.

**[Inadequate experiments]** There are some experiments to be done for verifying the effectiveness of the proposed method. (i) Training time comparisons with other methods and more comparisons for inference time in Table 10; (ii) It would be nice to see the performance by increasing each agent gradually. This can help better understand which agent is more effective and which one is useless. (iii) Ablation study conducted using VAC, LAC, or MAC individually, as well as in combinations of any two.

**[Disorganized reference format]** Please reformat the references, e.g., “Conference on Computer Vision and Pattern Recognition”, “Proceedings of the IEEE/CVF conference on computer vision and pattern recognition” and “Proceedings of the IEEE/CVF Conference on Computer Vision and Pattern Recognition (CVPR)”.

**Questions:**

Please refer to the weakness section.

**Limitations:**

Yes, they have provided the limitation section.

---

> ### Author Rebuttal · Authors · 2024-08-07
>
> Thanks for your constructive comments. We provide our feedbacks as follows.
>
> **Q1: The paper mentions that “it is the first unified transfer framework for generalizing vision-language foundation models with heterogeneous agent collaboration.” However, CaFo [67] had done similar things by adopting multiple heterogeneous agents. This statement may need a revision to differentiate from CaFo.**
>
> **A1:** Thanks for your suggestion.
> CaFo [67] leverages cascades of external models based on their output forms, which necessitates careful sequential arrangement of the models. This design inherently lacks **flexibility** due to the required model arrangement.
> In contrast, our TransAgent framework avoids this issue by employing a unified distillation process for heterogeneous agents, eliminating the complexity of sequential model arrangement. Additionally, TransAgent unloads all external models during inference, achieving **deployment efficiency** without the burden of a heavy model ensemble as seen with CaFo.
> Based on these distinctions, we will revise the statement to: “it is the first unified *distillation* framework for generalizing vision-language foundation models with *efficient* heterogeneous agent collaboration.”
>
> **Q2: Some works to be compared or discussed in related works are listed as follow:
> PromptKD (CVPR2024),
> DePT (CVPR2024),
> T\\(_{En}\\) (ICML2024),
> CoPrompt (ICLR2024),
> GraphAdapter (NeurIPS2023),
> Task Residual(CVPR2023).**
>
> **A2:** Thanks for this suggestion.
> We list the comparison on base-to-novel generalization of 11 benchmarks.
> (1) **PromptKD (CVPR2024)** also leverages distillation but uses the full training set of base and novel classes without labels. To align with the mainstream setting, we re-implemented its official code using only 16-shot labeled data. Our TransAgent clearly outperforms PromptKD under these conditions.
> (2) **T\\(_{En}\\) (ICML2024)** employs a model ensemble that must be maintained during inference, leading to inefficient deployment. In contrast, our TransAgent unloads all external models after training, with the inference pipeline identical to CLIP. This results in better performance with only **1/3** complexity during inference compared to **T\\(_{En}\\)**.
> (3) Compared to other prompt-learning based methods such as **DePT (CVPR2024)** and **CoPrompt (ICLR2024)**,
> our TransAgent simply achieves better performance.
> (4) For adapter-based methods such as **GraphAdapter (NeurIPS2023)** and **Task Residual(CVPR2023)**, we will refer them in the related works since the model and data settings are different.
>
> |Method|Training Data|Inference Model|Base Acc.|Novel Acc.|HM|
> |:-|:-|:-:|:-:|:-:|:-:|
> |PromptKD (CVPR2024)|16-shot + unlabeled|87M|86.96|80.73|83.73|
> |PromptKD (CVPR2024)|16-shot|87M|77.75|71.69|74.59|
> |T\\(_{En}\\) (ICML2024)|16-shot|268M|85.48|77.17|81.11|
> |DePT (CVPR2024)|16-shot|86M|85.19|76.17|80.43|
> |CoPrompt (ICLR2024)|16-shot|91M|84.00|77.23|80.48|
> |TransAgent (ours)|16-shot|86M|**85.29**|**77.62**|**81.27**|
>
> **Q3: It would be nice to see an advanced distillation with heterogeneous agents.**
>
> **A3:** Thanks for your insightful comments.
> To be noted,
> our primary goal is to introduce a **generic knowledge integration** method to enhance CLIP using heterogeneous agents. The main contribution lies in how we collaborate these heterogeneous agents to generate the knowledge vector for distillation, rather than the distillation operation itself.
> Specifically, we propose a novel **mixture-of-agent gating mechanism** (outlined in Equation 3, 5, and 7) for collaborating heterogeneous agents in each modality. This mechanism allows TransAgent to adaptively select and weight the knowledge from various agents, effectively reducing the introduction of irrelevant knowledge during distillation.
> The visualization of this process is provided in Figure 5, and the effectiveness of this design is demonstrated in the bottom sections of Tables 2-4 (Page 9). This innovative approach represents the advancement in our collaboration method.
>
> **Q4: More Experiments
> (i) Training time comparisons with other methods and more comparisons for inference time;
> (ii) It would be nice to see the performance by increasing each agent gradually.
> (iii) Ablation study conducted using VAC, LAC, or MAC individually, as well as in combinations of any two.**
>
> **A4:** (i) Since PromptKD (CVPR2024) also leverages distillation. We further make the comparison with it. The cost comparison on ImageNet is presented below, where all the models are trained for 5 epochs under 16-shot setting on a single A6000 GPU.
> Clearly, our model achieves a better performance with higher efficiency.
> |Method|Training Time (min)|Inference Time (sec)|HM|
> |:-|:-:|:-:|:-:|
> |PromptKD|146|201|71.29|
> |TransAgent|72|54|73.93|
>
> (ii) Note that we have already conducted the experiment of **whether use** a specific agent can help improve the performance in the second row of Table 2-4. The visualization of the gating weights in Figure 5 may also provide insights of which agent is more useful for a specific dataset. For instance, as for vision agents, DINO experts in recognizing general objects while lags behind on some fine-grained datasets (e.g., EuroSAT), where the other agents like SAM which focus more on details perform better.
>
> (iii) The results are presented below. As observed, each individual module is beneficial for improving the generalization ability of the foundation models. And the combinations of these modules further boost the results.
>
> |Module|Base Acc.|Novel Acc.|HM|\\(\Delta\\)|
> |:-|:-:|:-:|:-:|:-:|
> |baseline|84.21|71.79|77.51|-|
> |VAC|84.96|73.90|79.04|+1.53|
> |LAC|85.23|75.20|79.90|+2.39|
> |MAC|85.04|74.85|79.61|+2.10|
> |VAC + LAC|85.31|75.36|80.02|+2.51|
> |VAC + MAC|85.11|75.10|79.79|+2.28|
> |LAC + MAC|85.56|75.84|80.40|+2.89|
> |TransAgent|85.29|77.62|81.27|**+3.76**|
>
> **Q5: Disorganized reference format.**
>
> **A5:** Thanks for your advice. We will check and reformat the references in our final version.

---

> > ### Comment · Reviewer_DUuV · 2024-08-13
> >
> > Thanks for the response of authors. It has addressed my concerns, and I would like to raise my score to 6.

---

> > > ### Author Response · Authors · 2024-08-13
> > >
> > > Thanks for your feedback. We are glad to have addressed your concerns and receive your acknowledgement.

---

### Official Review · Reviewer_A3Ab · 2024-07-12

**Soundness:** 3
**Presentation:** 3
**Contribution:** 2
**Rating:** 5
**Confidence:** 2

**Summary:**

The paper introduces TransAgent, a novel framework designed to enhance vision-language foundation models like CLIP through the integration of knowledge from diverse, pre-trained expert models. These experts, which include vision, language, and multi-modal models, possess rich knowledge acquired from different modalities, tasks, networks, and datasets. TransAgent addresses the challenge of generalizing these models to new domains, especially under low-shot conditions, by proposing a unified transfer framework that leverages heterogeneous agent collaboration. The framework employs a mixture-of-agents gating mechanism to adaptively integrate external knowledge and uses multi-source distillation to transfer this knowledge into CLIP, enhancing its generalization ability without additional inference cost. Experiments demonstrate TransAgent's state-of-the-art performance on 11 visual recognition datasets, significantly outperforming methods like CoOp under low-shot settings and showing remarkable results on EuroSAT with large domain shifts. The paper's contributions lie in its innovative approach to knowledge transfer, the flexibility of its framework, and its significant improvements in model generalizability and efficiency.

**Strengths:**

The paper presents a highly original approach to enhancing vision-language foundation models by proposing the TransAgent framework. This framework innovatively integrates knowledge from a diverse set of pre-trained expert models into a unified system.

The use of a mixture-of-agents gating mechanism and multi-source distillation is a unique contribution that sets this work apart from existing methods.

The authors have conducted extensive experiments on 11 visual recognition datasets, demonstrating the effectiveness of TransAgent under various low-shot scenarios. The comparison with state-of-the-art methods like CoOp and the detailed ablation studies provide a solid understanding of the framework's capabilities and the impact of different components.

The paper is well-structured, with a clear abstract and introduction that succinctly summarize the contributions and scope of the work.

**Weaknesses:**

After changing the expert agent selection, the model needs to be retrained, and this structure is not plug-and-play.

The use of integrated features from multiple teacher models as supervision for distillation is widely used in other fields, and the authors have transferred this approach to the domain of this paper, which lacks sufficient novelty

**Questions:**

Could the authors provide insights into how the TransAgent framework generalizes to domains outside of the tested benchmarks, especially those with significantly different characteristics?

The paper mentions the potential introduction of irrelevant knowledge during distillation. What strategies are in place or could be considered to mitigate this issue?

The role of learnable prompts seems pivotal. How were the prompts engineered, and what impact did they have on the model's performance?

**Limitations:**

The authors have adequately addressed the limitations.

---

> ### Author Rebuttal · Authors · 2024-08-07
>
> Thanks for your constructive comments. We provide our feedbacks as follows.
>
> **Q1: After changing the expert agent selection, the model needs to be retrained, and this structure is not plug-and-play.**
>
> **A1:** We would like to clarify that the training effort required by our framework is minimal. All pre-trained models (including CLIP and the 11 agents) are frozen. Fine-tuning is performed on the learnable prompts and the gating mechanism, which requires only a small amount of computational effort. For example, under the 16-shot setting, training for 20 epochs on a single A6000 GPU takes less than 20 minutes for most benchmarks.
> This efficient process means that our framework can be **quickly adapted** to changes in expert agent selection or the testing of new downstream datasets.
> Additionally,
> our method is actually **'plug-and-play'**,
> i.e.,
> this concise collaboration framework can be integrated into various prompt-learning based methods with ease.
> We will further investigate this in future work.
>
> **Q2: The authors have transferred the distillation approach to the domain of this paper, which lacks sufficient novelty.**
>
> **A2:** To be noted,
> how to perform multi-source distillation is not trivial, since how to **extract and collaborate** knowledge from various heterogeneous teachers has not been explored for CLIP-like foundation models.
> We would like to emphasize the distinct technical novelty and contributions of our approach, specifically in the context of *Transfer Flexibility* (Introduction, Lines 49-55).
>
> (1) **Generic Knowledge Extraction Method:**
> Our method introduces a novel approach for extracting knowledge from heterogeneous agents, particularly the multi-modal ones. We design a unique method to extract prediction score vectors as multi-modal knowledge. This involves elaborate mining of vision-language alignment within these models. The efficacy is demonstrated in Table 4 (Page 9).
>
> (2) **Flexible Knowledge Collaboration Mechanism:**
> We propose a mixture-of-agent gating mechanism to integrate external knowledge from different agents. This mechanism allows TransAgent to dynamically select agents via soft weighting, enabling it to adapt effectively to few-shot settings across various target datasets. The effectiveness is detailed in Table 2-4 (Page 9).
>
> Beyond these technical innovations, TransAgent offers two additional significant advantages.
> (1) *Knowledge Versatility* (Line 45-48):
> To our best knowledge,
> TransAgent is **the first framework** to enhance CLIP-like models comprehensively using 11 heterogeneous agents, covering a wide range of vision, language, and multi-modal research areas.
> (2) *Deployment Efficiency* (Line 56-60):
> TransAgent allows all external agents to be unloaded after distillation. This guarantees the inference pipeline of the enhanced CLIP to be consistent with the original one, without the need for a cumbersome model ensemble, as evidenced in Table 10 (Page 16).
>
> **Q3: Could the authors provide insights into how the TransAgent framework generalizes to domains outside of the tested benchmarks, especially those with significantly different characteristics?**
>
> **A3:** The generalization capacity of our TransAgent framework is primarily credited to the **integration of diversified knowledge** from a wide range of heterogeneous agents that are pre-trained on different modalities, tasks, networks, and datasets.
> To substantiate this claim, we follow [1-5] to verify TransAgent on 11 downstream datasets spanning diverse domains arranging from image to video task, from natural to satellite images, from object to scene understanding, from regular to fine-grained recognition. The remarkable performance across these datasets demonstrates that TransAgent can **generalize effectively**, even when faced with **significant domain shifts**.
> We will investigate more diverse domain data in future work.
>
> **Q4: The paper mentions the potential introduction of irrelevant knowledge during distillation. What strategies are in place or could be considered to mitigate this issue?**
>
> **A4:** Thanks for your concern.
> To mitigate the introduction of irrelevant knowledge during distillation, we employ a **mixture-of-agent gating mechanism** (outlined in Equation 3, 5, and 7). This mechanism enables our TransAgent to adaptively select and weight the knowledge from different agents, thereby reducing the impact of irrelevant information.
> The visualization of this process is provided in Figure 5, and the effectiveness of this design is demonstrated in the bottom sections of Tables 2-4 (Page 9). This adaptive selection ensures that only the most relevant knowledge is distilled, enhancing the overall performance and robustness of the model.
>
> **Q5: The role of learnable prompts seems pivotal. How were the prompts engineered, and what impact did they have on the model's performance?**
>
> **A5:** To avoid the complex design of prompt engineering, we simply leverage the widely-used CoOP [1] method with a number of learnable prompt vectors, as mentioned in Lines 102-110 in the paper. Our method works well with such a simple prompting method, showing its effectiveness and flexibility. The importance of the learnable prompts as well as the prompt design have been studied thoroughly by previous works [3, 4, 5], and we simply follow the de-facto standard.
>
> **References.**
>
> [1] Learning to prompt for vision-language models. (IJCV-22)
>
> [2] Prompt, generate, then cache: Cascade of foundation models makes strong few-shot learners. (CVPR-23)
>
> [3] Maple: Multi-modal prompt learning. (CVPR-23)
>
> [4] Self-regulating prompts: Foundational model adaptation without forgetting. (ICCV-23)
>
> [5] Read-only prompt optimization for vision-language few-shot learning. (ICCV-23)

---

### Official Review · Reviewer_wEnm · 2024-07-18

**Soundness:** 3
**Presentation:** 2
**Contribution:** 2
**Rating:** 3
**Confidence:** 4

**Summary:**

The paper focuses on the challenge of vision-language foundation models (e.g., CLIP) struggling to generalize to diverse target domain data in downstream tasks. It highlights the potential of using expert models, which are pre-trained on various modalities, tasks, networks, and datasets, to improve generalization. The proposed TransAgent framework integrates the knowledge of these isolated expert models through a unified approach, enhancing CLIP's performance via multi-source knowledge distillation. TransAgent collaborates with 11 heterogeneous agents to empower vision-language models without adding inference phase costs. The framework achieves good performance on some visual recognition datasets.

**Strengths:**

1. The problem is of practical importance.
2. The experiments are sufficient.
3. The gains of knowledge distillation seem strong.

**Weaknesses:**

1. In my opinion, the proposed method leverages multiple pretty strong 'experts' for knowledge distillation, e.g., SAM, MAE, DINO, ViTDet, GPT, and Vicuna. However, almost all the baselines to be compared do not rely on external models. Hence, I think the majority of comparisons (e.g., Table 1 and Figure 4) may be unfair. The proper baselines should be other distillation methods (using the same external models as this paper), which the proposed method is actually orthogonal to most of the current baselines.

2. On top of 1, I think that if some significant technical contributions regarding knowledge distillation are not proposed, simply 'applying knowledge distillation methods to CLIP' may not be an acceptable novel contribution for me.

3. Given that multiple pretty strong 'experts' (e.g., SAM, MAE, DINO, ViTDet, GPT, and Vicuna) for knowledge distillation have been employed, the current gains of performance seem limited.

**Questions:**

Please refer to 'Weaknesses'.

**Limitations:**

The authors have addressed the limitations and potential negative societal impacts of their work.

---

> ### Author Rebuttal · Authors · 2024-08-07
>
> Thanks for your constructive comments. We provide our feedbacks as follows.
>
> **Q1: Almost all the baselines to be compared do not rely on external models. Hence, I think the majority of comparisons (e.g., Table 1 and Figure 4) may be unfair.**
>
> **A1:** Thank you for raising the concern.
>
> (1) Table 1 and Figure 4 are provided for the state-of-the-art (SOTA) comparison, rather than an ablation study focusing solely on the distillation method. Our aim is to identify **the most effective method**, regardless of the specific techniques employed. The results demonstrate that TransAgent achieves superior performance, highlighting its advancements in few-shot generalization.
>
> (2) During the inference phase, we unload all external models, ensuring that the inference process of TransAgent is identical to that of the original CLIP with learnable prompts. Therefore, it is fair to compare our method with others using learnable prompts, as they share **the same inference setting**, as presented in Table 1.
>
> (3) Our TransAgent also outperforms CaFo, which similarly relies on external models, as shown in Figure 4. This comparison further underscores the effectiveness and fairness of our approach.
>
> **Q2: The proper baselines should be other distillation methods (using the same external models as this paper), which the proposed method is actually orthogonal to most of the current baselines.**
>
> **A2:** Thanks for your insightful feedback.
>
> (1) We have included baselines that involve distillation with external models. As indicated in the second row of Tables 2-4 (Page 9), the use of distillation from each external model contributes to performance improvement. Our proposed heterogeneous agent collaboration (the Gating results shown in Tables 2-4) consistently achieves the best results. This fairly demonstrates the effectiveness of our method in leveraging multiple external models.
>
> (2) For the other distillation method in the research of CLIP [PromptKD, CVPR2024], it is not feasible to use the exact same set of external models as in our study due to differences in frameworks and settings. Specifically, the referenced method employs a transductive setting, which differs from our approach. Nonetheless, we have included this method as a state-of-the-art (SOTA) baseline for comparison.
> For instance, on the EuroSAT benchmark, which involves a **significant domain shift** with satellite images, our method achieves an accuracy of 83.43 on novel classes, compared to 82.08 for PromptKD. Notably, we achieve this with only 16-shot labeled data in the base class, whereas PromptKD utilizes 16-shot labeled data in the base class and the full training set without labels in the base and novel classes. This demonstrates our method's superior performance.
>
> **Q3: If some significant technical contributions regarding knowledge distillation are not proposed, simply 'applying knowledge distillation methods to CLIP' may not be an acceptable novel contribution.**
>
> **A3:** To be noted, how to perform multi-source distillation is not trivial, since how to **extract and collaborate** knowledge from various heterogeneous teachers has not been explored for CLIP-like foundation models.
> We would like to emphasize the distinct technical novelty and contributions of our approach, specifically in the context of *Transfer Flexibility* (Introduction, Lines 49-55).
>
> (1) **Generic Knowledge Extraction Method:**
> Our method introduces a novel approach for extracting knowledge from heterogeneous agents, particularly the multi-modal ones. We design a unique method to extract prediction score vectors as multi-modal knowledge. This involves elaborate mining of vision-language alignment within these models. The efficacy is demonstrated in Table 4 (Page 9).
>
> (2) **Flexible Knowledge Collaboration Mechanism:**
> We propose a mixture-of-agent gating mechanism to integrate external knowledge from different agents. This mechanism allows TransAgent to dynamically select agents via soft weighting, enabling it to adapt effectively to few-shot settings across various target datasets. The effectiveness is detailed in Table 2-4 (Page 9).
>
> Beyond these technical innovations, TransAgent offers two additional significant advantages.
> (1) *Knowledge Versatility* (Line 45-48):
> To our best knowledge,
> TransAgent is **the first framework** to enhance CLIP-like models comprehensively using 11 heterogeneous agents, covering a wide range of vision, language, and multi-modal research areas.
> (2) *Deployment Efficiency* (Line 56-60):
> TransAgent allows all external agents to be unloaded after distillation. This guarantees the inference pipeline of the enhanced CLIP to be consistent with the original one, without the need for a cumbersome model ensemble, as evidenced in Table 10 (Page 16).
>
> **Q4: Given that multiple pretty strong `experts' for knowledge distillation have been employed, the current gains of performance seem limited.**
>
> **A4:** Thank you for raising this concern. The performance gains achieved by our TransAgent are indeed significant, as highlighted by the following observations. *First*, TransAgent operates similarly to CoOp [2] during inference, utilizing learnable prompts while unloading all external models.
> With the same inference pipeline, TransAgent significantly outperforms the popular CoOp by around **10%** on average, and by as much as **20%** on the EuroSAT dataset which contains large domain shifts, under the same low-shot setting in Table 1. *Second*, when compared to CaFo [3], which also utilizes external models, TransAgent achieves approximately **5%** higher accuracy on average, while using only **1/10** of the inference time as shown in Table 10.
>
> **References.**
>
> [1] PromptKD: Unsupervised Prompt Distillation for Vision-Language Models (CVPR-24)
>
> [2] Learning to prompt for vision-language models. (IJCV-22)
>
> [3] Prompt, generate, then cache: Cascade of foundation models makes strong few-shot learners. (CVPR-23)

---

### Official Review · Reviewer_gY7M · 2024-07-20

**Soundness:** 3
**Presentation:** 3
**Contribution:** 2
**Rating:** 5
**Confidence:** 4

**Summary:**

This paper aims to handle heterogeneous foundation model combination across different pretrained backbones. Instead of using vanilla ensemble, it proposes to use a distillation process to transfer knowledges from different agents. More specifically, it uses a learnable gate module to integrate different knowledge sources. Extensive experiments show the propose method outperforms other related models.

**Strengths:**

1. Efficiently combining several pretrained foundation models is a valuable research direction.
2. Overall, the writing is clear to read and follow, such as motivation, methodology, and empirical results.
3. Compared with baseline, the proposed model does not need to involve extra inference cost, benefitted by a distillation process.
4. Empirical results show the model superiority.

**Weaknesses:**

1. Overall, I think this is more like a jointly distillation technique, instead of so-called agent collaboration, which may lead to certain misunderstanding.
2. Some figures are a little confused to understand. A clean and informative figures to show the whole working pipeline will be helpful.
3. Even if the empirical results are good, the proposed method still lacks of research novelty.

**Questions:**

See the weaknesses section above.

---

> ### Author Rebuttal · Authors · 2024-08-07
>
> Thanks for your constructive comments. We provide our feedbacks as follows.
>
> **Q1: The proposed method is more like a jointly distillation technique, instead of so-called agent collaboration, which may lead to certain misunderstanding.**
>
> **A1:** We would like to clarify this misunderstanding. Joint distillation is our goal, while agent collaboration is our method to achieve this goal. Specifically, we introduce a generic gating mechanism for each modality (outlined in Equation 3, 5 and 7). This mechanism allows us to flexibly weight the contributions of various teacher models, thereby creating a summarized knowledge vector that is subsequently used for distillation within the given modality. We describe this process as **"agent collaboration"** because it involves all the agents (models) working together adaptively towards a shared objective, i.e., constructing a knowledge vector for effective distillation. This collaborative aspect distinguishes our method.
> We will clarify this in the revision.
>
> **Q2: Some figures are a little confused to understand. A clean and informative figures to show the whole working pipeline will be helpful.**
>
> **A2:** Thanks for your suggestions. Figure 1 actually shows the whole pipeline of our TransAgent, where
> we extract external knowledge from heterogeneous models in each modality and leverage the knowledge of all the modalities to boost vision-language foundation models. The current figures mainly focuse on "what" knowledge are integrated. We will add more informative figures to show "how" to integrate these knowledge in the revision.
>
> **Q3: Even if the empirical results are good, the proposed method still lacks of research novelty.**
>
> **A3:** To be noted,
> how to perform multi-source distillation is not trivial, since how to **extract and collaborate** knowledge from various heterogeneous teachers has not been explored for CLIP-like foundation models.
> We would like to emphasize the distinct technical novelty and contributions of our approach, specifically in the context of *Transfer Flexibility* (Introduction, Lines 49-55).
>
> (1) **Generic Knowledge Extraction Method:**
> Our method introduces a novel approach for extracting knowledge from heterogeneous agents, particularly the multi-modal ones. We design a unique method to extract prediction score vectors as multi-modal knowledge. This involves elaborate mining of vision-language alignment within these models. The efficacy is demonstrated in Table 4 (Page 9).
>
> (2) **Flexible Knowledge Collaboration Mechanism:**
> We propose a mixture-of-agent gating mechanism to integrate external knowledge from different agents. This mechanism allows TransAgent to dynamically select agents via soft weighting, enabling it to adapt effectively to few-shot settings across various target datasets. The effectiveness is detailed in Table 2-4 (Page 9).
>
> Beyond these technical innovations, TransAgent offers two additional significant advantages. (1) *Knowledge Versatility* (Line 45-48):
> To our best knowledge, TransAgent is **the first framework** to enhance CLIP-like models comprehensively using 11 heterogeneous agents, covering a wide range of vision, language, and multi-modal research areas. (2) *Deployment Efficiency* (Line 56-60):
> TransAgent allows all external agents to be unloaded after distillation. This guarantees the inference pipeline of the enhanced CLIP to be consistent with the original one, without the need for a cumbersome model ensemble, as evidenced in Table 10 (Page 16).

---

### Official Review · Reviewer_QaNB · 2024-07-26

**Soundness:** 4
**Presentation:** 3
**Contribution:** 3
**Rating:** 5
**Confidence:** 4

**Summary:**

This paper introduces a TransAgent framework, which guides CLIP to generalize with multi-source knowledge distillation. The framework contains three kinds of collaboration, including vision models, language models and muti-modal models. A Mixture-of-Agents (MoA) gating mechanism is proposed to adaptively integrate the knowledge. SOTA is achieved on 11 datasets under the few-shot scenarios.

**Strengths:**

This method guarantees the deployment efficiency, as knowledge from heterogeneous agents are distillated and injected into CLIP.
This framework can be extended flexibly. Any expert can be introduced as a teacher model.
The paper is easy to follow. Figures and experiments are clearly presented.

**Weaknesses:**

The method applies multi-teacher distillation in clip with prompts, and popular large models are exploited as teachers. Thus, the novelty is limited.
Given that the framework comprises three components — VAC, LAC, and MAC—the reviewer insists that the  results of every module (only-VAC, only-LAC, only-MAC) will be more persuasive to demonstrate their respective effectiveness (i.e. only-VAC's, only-LAC's, only-MAC's).
The clip absorbs various knowledge from 11 excellent agents. In addition to few-shot learning on 11 downstream datasets, experiments on general dataset are also encouraged, like WebQA and CIRR.
In the related works, research on muti-teacher distillation should be included. Baselines in the experiments should be introduced in the related work.
Typos in line 259: knowledege -> knowledge

**Questions:**

11 agents contain different domain knowledges. Irrelevant information or inconsistent information may be induced. Is there any challenge in the training and how to deal with challenges?
How to prevent overfitting when distilled from 11 pretrained models?
The λ_2 is far greater than other λ values, does it mean that the loss of LAC is more important than other losses?

**Limitations:**

The authors have listed the limitation in the paper. It is encouraged to provide more analysis and solutions.

---

> ### Author Rebuttal · Authors · 2024-08-07
>
> Thanks for your constructive comments. We provide our feedbacks as follows.
>
> **Q1: The method applies multi-teacher distillation in clip with prompts, and popular large models are exploited as teachers. Thus, the novelty is limited.**
>
> **A1:** To be noted, how to perform multi-source distillation is not trivial, since how to **extract and collaborate** knowledge from various heterogeneous teachers has not been explored for CLIP-like foundation models. We would like to emphasize the distinct technical novelty and contributions of our approach, specifically in the context of *Transfer Flexibility* (Introduction, Lines 49-55).
>
> (1) **Generic Knowledge Extraction Method:** Our method introduces a novel approach for extracting knowledge from heterogeneous agents, particularly the multi-modal ones. We design a unique method to extract prediction score vectors as multi-modal knowledge. This involves elaborate mining of vision-language alignment within these models. The efficacy is demonstrated in Table 4 (Page 9).
>
> (2) **Flexible Knowledge Collaboration Mechanism:** We propose a mixture-of-agent gating mechanism to integrate external knowledge from different agents. This mechanism allows TransAgent to dynamically select agents via soft weighting, enabling it to adapt effectively to few-shot settings across various target datasets. The effectiveness is detailed in Table 2-4 (Page 9).
>
> Beyond these technical innovations, TransAgent offers two additional significant advantages. (1) *Knowledge Versatility* (Line 45-48): To our best knowledge, TransAgent is **the first framework** to enhance CLIP-like models comprehensively using 11 heterogeneous agents, covering a wide range of vision, language, and multi-modal research areas. (2) *Deployment Efficiency* (Line 56-60):
> TransAgent allows all external agents to be unloaded after distillation. This guarantees the inference pipeline of the enhanced CLIP to be consistent with the original one, without the need for a cumbersome model ensemble, as evidenced in Table 10 (Page 16).
>
> **Q2: The results of every module (only-VAC, only-LAC, only-MAC) will be more persuasive to demonstrate their respective effectiveness.**
>
> **A2:** The ablation is conducted below. As observed in the table, each module is effective and their combination further boosts the overall performance on 11 benchmarks (the same setting as Table 1 in the paper).
>
> |Module|Base Acc.|Novel Acc.|HM|
> |:-|:-:|:-:|:-:|
> |baseline|84.21|71.79|77.51|
> |VAC|84.96|73.90|79.04|
> |LAC|85.23|75.20|79.90|
> |MAC|85.01|74.85|79.61|
> |TransAgent|85.29|77.62|81.27|
>
> **Q3: In addition to few-shot learning on 11 downstream datasets, experiments on general dataset are also encouraged, like WebQA and CIRR.**
>
> **A3:** Note that, these 11 downstream datasets have been widely used to evaluate the generalization capacity of foundation models [1-4], since they are actually the general datasets arranging from image to video task, from natural to satellite images,
> from object to scene understanding, from regular to fine-grained recognition. Hence, we follow the mainstream setting for evaluation.
>
> **Q4: In the related works, research on muti-teacher distillation should be included. Baselines in the experiments should be introduced
> in the related work. Typos in line 259: knowledege -> knowledge.**
>
> **A4:** Thanks for your suggestions on related works. We will include baseline methods and research on multi-teacher distillation [5-7] in the related work in our final version. Additionally, we will fix the typos in the revision.
>
> **Q5: 11 agents contain different domain knowledge.
> Irrelevant information or inconsistent information may be induced.
> (1) Is there any challenge in the training and how to deal with challenges?
> (2) How to prevent overfitting when distilled from 11 pretrained models?
> (3) The \\(\lambda_2\\) is far greater than other \\(\lambda\\) values, does it mean that the loss of LAC is more important than other losses?**
>
> **A5:** Thanks for raising the concerns.
>
> (1) The primary challenge lies in the varying importance of the 11 agents for different target datasets. Treating all the knowledge equally would indeed hinder generalization. To address this, we implement a **mixture-of-agent gating mechanism** in the few-shot training. This mechanism enables us to adaptively weight the contributions of the agents for different target datasets. The visualization is provided in Figure 5, and its effectiveness is demonstrated at the bottom of Tables 2-4 (Page 9).
>
> (2) We mitigate the risk of overfitting, since all pre-trained models (including CLIP and 11 agents) are frozen during the training process. Fine-tuning primarily works on a few learnable prompts and the gating mechanism, which involves minimal adjustments (conducted over 20 epochs).
>
> (3) The main reason is that, the absolute value of LAC loss is smaller than other losses. To ensure that the LAC loss contributes effectively to the training process, we choose a greater \\(\lambda_2\\). This adjustment ensures that the weighted values of all the losses are comparable, and be effective when training our TransAgent. We ablate the value of \\(\lambda_2\\) below.
>
> |\\(\lambda_2\\)|Base|Novel|HM|
> |:-:|:-:|:-:|:-:|
> |1.0|84.89|74.36|79.28|
> |10.0|84.75|75.47|79.84|
> |20.0|84.97|76.50|80.51|
> |25.0|85.29|77.62|81.27|
> |30.0|85.15|77.31|81.04|
>
> **References.**
>
> [1] Learning to prompt for vision-language models. (IJCV-22)
>
> [2] Prompt, generate, then cache: Cascade of foundation models makes strong few-shot learners. (CVPR-23)
>
> [3] Maple: Multi-modal prompt learning. (CVPR-23)
>
> [4] Self-regulating prompts: Foundational model adaptation without forgetting. (ICCV-23)
>
> [5] Mitigating Accuracy-Robustness Trade-Off Via Balanced Multi-Teacher Adversarial Distillation. (TPAMI-24)
>
> [6] Let All be Whitened: Multi-teacher Distillation for Efficient Visual Retrieval. (AAAI-24)
>
> [7] AM-RADIO: Agglomerative Vision Foundation Model Reduce All Domains Into One. (CVPR-24)

---

### Author Rebuttal · Authors · 2024-08-07

We sincerely thank all the reviewers for their constructive comments.
We are delighted to receive positive feedback such as
"the idea is intriguing" (**DUuV**),
"valuable research direction" (**gY7M**),
"can be extented flexibly" (**QaNB**),
"unique contribution" (**A3Ab**),
"of practical importance" (**wEnm**),
"figures are clearly presented" (**DUuV, QaNB**),
"easy to follow" (**QaNB, gY7M, DUuV**)
and "well-structured" (**A3Ab, DUuV, gY7M**).
We have carefully addressed all the concerns raised by the reviewers in the individual response section.

---

### Decision · Program_Chairs · 2024-09-25

**Decision:**

Accept (poster)

**Comment:**

This paper introduces a unified framework that utilizes various agents to integrate their knowledge for enhancing CLIP-like models. The framework is flexible, achieving superior results compared to state-of-the-art approaches, without incurring additional inference costs compared to the baseline. The writing is clear and easy to follow, and the authors have provided a thorough rebuttal addressing the reviewers' questions and concerns.